# Detecting of the Crack and Leakage in the Joint of Precast Concrete Segmental Bridge Using Piezoceramic Based Smart Aggregate

**DOI:** 10.3390/s20185398

**Published:** 2020-09-21

**Authors:** Jianqun Wang, Zhe Fan

**Affiliations:** 1Hunan Provincial Key Laboratory of Structures for Wind Resistance and Vibration Control, School of Civil Engineering, Hunan University of Science and Technology, Xiangtan 411201, China; 2Department of Mechanical Engineering, University of Houston, Houston, TX 77004, USA; noodles0305@163.com; 3Institute of Water Resources and Hydropower Research, Beijing 100038, China

**Keywords:** structural health monitoring (SHM), piezoceramic transducers, smart aggregate (SA), precast concrete segmental bridge (PCSB), detection of crack and leakage

## Abstract

Precast concrete segmental bridges (PCSBs) have been widely used in bridge engineering due to their numerous competitive advantages. The structural behavior and health status of PCSBs largely depend on the performance of the joint between the assembled segments. However, due to construction errors and dynamic loading conditions, some cracks and leakages have been found at the epoxy joints of PCSBs during the construction or operation stage. These defects will affect the joint quality, negatively impacting the safety and durability of the bridge. A structural health monitoring (SHM) method using active sensing with a piezoceramic-based smart aggregate (SA) to detect the crack and leakage in the epoxy joint of PCSBs was proposed and the feasibility was studied by experiment in the present work. Two concrete prisms were prefabricated with installed SAs and assembled with epoxy joint. An initial defect was simulated by leaving a 3-cm crack at the center of the joint without epoxy. With a total of 13 test cases and the different lengths of cracks without water and filled with water were simulated and tested. Time-domain analysis, frequency-domain analysis and wavelet-packet-based energy index (WPEI) analysis were conducted to evaluate the health condition of the structure. By comparing the collected voltage signals, Power Spectrum Density (PSD) energy and WPEIs under different healthy states, it is shown that the test results are closely related to the length of the crack and the leakage in the epoxy joint. It is demonstrated that the devised approach has certain application value in detecting the crack and leakage in the joint of PCSBs.

## 1. Introduction

The precast segmental construction (PSC) method is very suitable for medium- and long-span bridge projects due to its efficiency and economy [1,2,3]. By precasting the segmental units in the strictly controlled factory environment, high-quality structures with precise dimensions, desired strengths and reduced shrinkage and creep effects can be fabricated [4,5,6]. Moreover, there is little impact on the surrounding environment since the temporary supports and large amounts of falseworks are usually completely eliminated [7,8]. Since the Choisy-le-Roi bridge, the first precast concrete segmental bridge (PCSB), was built in France in 1962, great progress has been made in the precast segmental construction method and thousands of bridges have been built with this method [9,10,11]. In these PCSBs, epoxy joints are usually adopted because of the excellent material mechanical properties [12,13,14]. However, due to construction errors, such as uneven epoxy, inadequate bonding and tiny cracks, worsened by bridge vibrations [15,16], there will be defects at the joints. Some cracking and leakage has been found at the joints of PCSBs during the construction or operation stage. As illustrated in Figure 1, during the field investigation, the authors found that there were cracks and leakage in the top flange and web of a newly constructed precast concrete segmental box girder bridge. These defects will affect the joint quality, negatively impacting the safety and durability of the bridge [17,18,19]. Therefore, it is very necessary to detect and monitor the crack and leakage in the joint of PCSBs. 

Inspection via non-destructive methods is a choice to detect these defects mentioned above based on a fixed time interval [20,21,22]. To provide real-time information on structural well being, structural health monitoring (SHM) has received much attention recently [23,24,25]. In practical engineering, the commonly used methods for detecting the cracks in the concrete include image-based methods [26,27,28], radar-based methods [29,30], ultrasonic [31,32,33], and impact echo (IE) [34,35], among others. In recent years, studies show that the lead zirconate titanate (PZT) has been applied in SHM [36,37] due to its superiorities of low cost, strong piezoelectric effect [38,39], sensing and actuation [40,41], wide bandwidth [42,43] and energy harvesting [44,45]. Using active sensing methods, PZTs have been successfully applied in monitoring concrete bending stiffness [46], detecting the grouting quality in prestressed tendon duct [47], inspecting the loosened connection [48] and characterizing cracks [49,50]. With the development of smart material, PZTs in the form of smart aggregates (SAs) have emerged as a new tool to conduct SHM of concrete structures due to its advantages of embeddability and durability [51,52]. Xu et al. used SA to detect the interface debonding of a concrete-filled steel tube with wavelet packet analysis and provided an innovative method to inspect the debonding damage in a concrete-filled steel tube [53]. Wu et al. inspected the interlayer slide with an active sensing approach [54]. Liu et al. monitored the influence of axial loads on the health of concrete structures using embedded SAs and provided a new method to evaluate the stress status in concrete structures [55].

The feasibility study of the PZT-based approach in detecting the crack or leakage has also been reported in some research. Zhu et al. reported an innovative approach to detect the leakage of a gas pipeline with PZT sensors and verified the effectiveness in a 55-m-long pipeline [56]. Du et al. conducted an active sensing approach to identify the crack damage in the pipeline using a damage index matrix with PZT [57]. Liu et al. monitored the water seepage in concrete columns with piezoceramic-based SAs [58]. Feng et al. proposed a reliable approach to inspect the circumferential and axial cracks and the leakage in a reinforced concrete pipe [59]. Kong et al. successfully developed a method to detect water presence in the crack of a concrete beam specimen with SAs [60]. 

However, little or no literature on detecting the crack and leakage in the joint of PCSBs has been reported, to the authors’ best knowledge. Moreover, there are few studies to evaluate the length of the empty crack and water-filled crack based on the SHM method. In order to detect these defects, an SHM method using active sensing with piezoceramic-based SA is proposed. Two concrete specimens were prefabricated with an SA installed at the center and assembled with epoxy. An initial defect was simulated by leaving a 3-cm crack at the center of the joint without epoxy. A total of 13 test cases, by which the different lengths of cracks without water and filled with water were simulated, were tested in this experiment to detect the joint quality and leakage. Time-domain analysis, frequency-domain analysis and wavelet-packet-based energy index (WPEI) analysis were conducted to evaluate the health condition of the structure. Comparison of the collected voltage signals, Power Spectrum Density (PSD) energy and WPEIs under different healthy states show test results that are closely related to the length of the crack and the leakage in the epoxy joint. It is demonstrated that the devised approach has certain application value in detecting the crack and leakage in the joint of PCSBs.

## 2. Detection Principle

### 2.1. Piezoelectric Active Sensing Method

The stress wave-based active sensing approach was employed to detect the crack and leakage in the joint of PCSBs using piezoceramic-based SA. The experiment is designed based on field investigation and the schematic diagram is shown in Figure 2. Two concrete prisms were prefabricated with embedded SAs and assembled with an epoxy joint, leaving a length of 3 cm at the center of the joint without epoxy to simulate the initial defect. Based on the piezoelectricity properties, SA could be used as the actuator or sensor. In this experiment, SA1 was regarded as an actuator and SA2 as a sensor for collecting the stress wave signal. 

As demonstrated in Figure 2a, when there is no crack in the epoxy joint, the SA2 can collect the most signals, as the stress wave can directly transmit through the epoxy joint. As shown in Figure 2b, the initial crack in the epoxy joint will attenuate stress wave energy on the wave path, and the received energy will be smaller than the test case in Figure 2a. As illustrated in Figure 2c, the crack is filled with water functioning as a conduit. It can be inferred that the received energy of SA2 is more than that of the test case in Figure 2b, but less than the case in Figure 2a. The test case in Figure 2a involves no crack and is considered healthy in status, which can provide the baseline data. The received signal can be characterized using WPEI presented in the next section.

### 2.2. Wavelet Packet-Based Energy Index (WPEI)

WPEI has been used as a quantitative method to analyze the received signals and to evaluate the structural health [61,62,63]. The WPEI is sensitively correlated to the defect condition of the epoxy joint. Therefore, the received energy of SA2 can be used to describe the defect condition such as crack and leakage in the epoxy joint.

In the experiment, the excitation signal applied to S1 is a sweep sine wave from 1000 Hz to 200,000 Hz, and a wide frequency range of signals will be collected by S2. Then, the collected signal can be decomposed into a set of frequency bands, of which the signal energy can be calculated. Finally, by accumulating all the energies of the frequency bands, the total energy received by SA2 can be obtained.

Based on WPEI, in the energy analysis, the received signal $X_{i}$ can be decomposed into 2^n^ frequency sets by an n-level wavelet packet. Then, the decomposed signal $X_{i,j}$ can be expressed as
(1)$$X_{i,j}=\left[X_{i,j,1},X_{i,j,2},\cdots ,X_{i,j,m}\right]$$
where i and j are respectively the various time and frequency band $\left(j=1,2,3,\cdots ,2^{n}\right)$, and m represents the number of received sampling data by the SA. Therefore, the energy $E_{i,j}$ can be calculated by the decomposed signal $X_{i,j}$ and shown as follow,
(2)$$E_{i,j}={\left\|X_{i,j}\right\|}^{2}={X_{i,j,1}}^{2}+{X_{i,j,2}}^{2}+\cdots +{X_{i,j,m}}^{2}$$

The gross energy $E_{i}$ can be obtained and expressed as follows,
(3)$$E_{i}=\left[\begin{array}{ccccc} E_{i,1}, & E_{i,2}, & E_{i,3}, & \cdots & E_{i,2^{n}} \end{array}\right]$$

The WPEI defined by Equations (1)–(3) has been applied to signal processing in SHM of various structures, such as concrete structures [64,65], steel structures [66,67], composite structures [68,69] among others [70,71]. The received energy by SA can be calculated with WPEI in this experiment. The WPEI value of the specimen without any crack is defined as the initial value. It can be inferred that if there are some defects in the joint, the value will decrease.

## 3. Experimental Setup and Process

### 3.1. Specimen Design and Fabrication

The specimen was designed based on engineering investigation. As illustrated in Figure 3, two concrete prisms with the dimension of 100 × 100 × 200 mm were prefabricated to simulate two segmental concrete girders. The concrete mixture included Portland cement PO 42.5, natural sand, coarse aggregate and water. 

The concrete specimens were prefabricated with an SA installed at the center and cured for 10 days. Then, the specimens were assembled with epoxy. The thickness of the epoxy joint was 1.5 mm and strictly controlled. The initial defect was simulated by leaving a length of 3 cm at the center of the joint without epoxy. After being cured for another 3 days, the joint quality and leakage of the specimen were detected by an active sensing method based on SA. The material properties of the transducers and the epoxy were illustrated in Table 1.

### 3.2. Experimental Setup

The experimental setup is illustrated in Figure 4. The specimen included two installed SAs, two joined concrete prisms with epoxy, and the detecting device included a data acquisition system (NI USB-6363), a power amplifier and a supported laptop. In the experiment, a sweep sine wave was generated by the data acquisition system. Then, the power amplifier, with a gain of 50 for a piezoceramic load, was used to drive SA1 as an actuator.

### 3.3. Experimental Procedure

As illustrated in Table 2, a total of 13 test cases were designed in this experiment to detect the joint quality and leakage. The two concrete prisms were assembled, leaving a length of 3 cm at the center of the joint without epoxy to simulate the initial defect. Then, the joined specimen was tested as test case 1. In test case 2, the bottom of the crack was attached with waterproof tape, and fresh water was injected into the 3 cm crack by a syringe. The test was conducted when the crack was full of water. When the test was finished, the tape at the bottom was opened to let the water flow out completely. Cold air was used from a blow drier to blow the crack for 10 min to dry the residual water in the crack. Then, we used a hacksaw to expand the 3 cm crack by 1 cm on both sides along the epoxy joint, so that the length of the crack became 5 cm. Test cases 3 through 12 were conducted in the same way as the first and second cases. When test case 12 was finished, the crack was dried for 10 min with the blow drier, and then the specimen was left for two days to dry fully. Finally, the 13 cm crack of the specimen was filled with epoxy and cured for 3 days. Since test case 13 involved no crack, the structure was in healthy condition and the test result could be regarded as the baseline data.

In each test case, SA1, as an actuator, was excited by a sweep sine signal to generate the stress wave. SA2 received the stress wave transmitted through the epoxy joint. A special program developed by LabVIEW software was used for experimental testing. The detailed parameters of the swept sine wave signal are shown in Table 3. Based on the acquired data, the joint quality and leakage were evaluated in the following section.

## 4. Experimental Result and Analysis

### 4.1. Time-Domain Analysis

Time-domain signal was the original data acquired by SA2. By time-domain analysis, the trend of the received voltage signal can be obtained. Some representative voltage signals collected from different test cases are shown in Figure 5, Figure 6, Figure 7 and Figure 8. 

Figure 5 shows the comparison of the specimen without a crack (test case 13), 3 cm crack without water (test case 1) and 3 cm crack filled with water (test case 2). As can be seen from the figure, the shape of the signal changes as the healthy state of the specimen changes. However, it is easier to quantify the healthy state with maximal amplitudes of the signals. For test case 13, specimen without a crack, the aptitude of the received signal is between −0.071 V and 0.071 V. For test case 1 with 3 cm crack without water, the aptitude of the received signal is between −0.057 V and 0.057 V, decreasing abruptly from test case 13. For test case 2 with 3 cm crack filled with water, the aptitude is between −0.064 V and 0.064 V, increasing obviously from test case 1. It is demonstrated that the received aptitude will decrease for the crack in the epoxy joint will attenuate stress wave energy on the wave path. While the crack is filled with water, the aptitude will increase for the stress wave can transmit through the crack with the filled water which functions as a conduit. However, in this condition, the received aptitude is still less than test case 13 without a crack, for the stress wave conducts better in epoxy joint than in water.

Figure 6 shows the comparison of test case 3 (5 cm crack without water) and test case 4 (5 cm crack filled with water). As can be seen from the figure, for test case 3, the specimen with 5 cm empty crack, the aptitude of the received signal is between −0.045 V and 0.045 V, decreasing abruptly from test case 1 (3 cm crack without water). For test case 4 with 5 cm water-filled crack, the aptitude is between −0.052 V and 0.052 V, increasing obviously but still less than test cases 1 and 2. It is demonstrated that the received aptitude will decrease abruptly when the length of the crack in the epoxy joint increases to 5 cm.

Figure 7 shows the comparison of test case 7 (9 cm crack without water) and test case 8 (9 cm crack filled with water). As illustrated in the figure, for test case 7, the crack length is 9 cm without water, the aptitude of the received signal is between −0.037 V and 0.037 V, which is far less than test case 1 (3 cm crack without water). For test case 8, the crack length is 9 cm filled with water, and the aptitude of the received signal is between −0.041 V and 0.041 V, which is more than test case 7 but far less than test case 2 (3 cm crack filled with water). Therefore, it can be speculated that, as the length of the crack increases, the received aptitude decreases. Furthermore, when the crack length exceeds 9 cm, the voltage-increasing trend between the empty crack and water-filled crack is not obvious.

Figure 8 shows the comparison of test case 11 (13 cm crack without water) and test case 12 (13 cm crack filled with water). As can be seen from the figure, the attenuation trend of the tested amplitude is consistent with those in Figure 5, Figure 6 and Figure 7. 

In order to further study the influence of the length of the empty crack on the received signal, all the time-domain signals of the test cases with empty crack are summarized in a single figure, and compared with the test case without a crack, as shown in Figure 9. It is demonstrated that the received aptitude will decrease as the length of the empty crack increases.

On the other hand, all the time-domain signals of the test cases with cracks filled with water are summarized in Figure 10, compared with the test case with no crack. It is demonstrated that the received aptitude will decrease as the length of the water-filled crack increases.

### 4.2. Frequency-Domain Analysis

The frequency-domain analysis can be performed on the base of the original data detected by SA2 to evaluate the frequency sensitive range and the PSD energy of the structure. The damage level of structures can be determined by analyzing the attenuation degree of signals at different frequencies. The time-domain signals are transformed into frequency-domain signals by Fourier transform. The results of the frequency-domain analysis for some typical test cases are shown in Figure 11.

Figure 11a shows the comparison of PSD energy for the specimen without a crack (test case 13), 3 cm crack without water (test case 1) and 3 cm crack filled with water (test case 2). As can be seen from the figure, the shape of the frequency-domain signals changes as the healthy state of the specimen changes. However, it is easier to quantify the healthy state with maximal amplitudes of the signals. The sensitive frequency range is 50 kHz to 180 kHz for the sensor signal. For test case 13, test case 1 and test case 2, the maximum PSD energy is 8.92 × 10^−9^ V^2^/Hz, 6.12 × 10^−9^ V^2^/Hz and 6.95 × 10^−9^ V^2^/Hz, respectively. It is demonstrated that the PSD energy will decrease for the crack in the epoxy joint. Additionally, the energy will increase if the crack is filled with water. However, in this condition, the PSD energy is still less than test case 13 without a crack.

Figure 11b illustrates the comparison of PSD energy between test case 3 (5 cm crack without water) and test case 4 (5 cm crack filled with water). As can be seen from the figure, for test cases 3-4, the maximum PSD energy is 4.03 × 10^−9^ V^2^/Hz and 5.71 × 10^−9^ V^2^/Hz, respectively. It is demonstrated that the PSD energy decreases abruptly when the length of the crack in the epoxy joint increases to 5 cm.

As can be seen from Figure 11c,d, the attenuation trend of the PSD energy is consistent with those in Figure 11a,b.

In order to further study the influence of the length of the empty crack on the PSD energy, all the frequency-domain signals of the test cases with empty crack and no crack are summarized in a single figure, as shown in Figure 12a. It is demonstrated that the PSD energy decreases as the length of the empty crack increases. On the other hand, all the frequency-domain signals of the test cases with the water-filled crack and no crack are summarized in Figure 12b. It is demonstrated that the PSD energy decreases as the length of the water-filled crack increases. 

As can be seen from Figure 11 and Figure 12, it is demonstrated that 50 kHz to 180 kHz is the sensitive frequency range for the sensor signal. As the length of the empty crack increases, PSD energy will decrease. For a certain length of the crack, the PSD energy of the water-filled crack has a certain degree of growth, compared with an empty crack. Furthermore, as the length of the water-filled crack increases, PSD energy decreases.

### 4.3. Wavelet Packet Energy

Based on the collected signal, the WPEI analysis can be conducted to assess the health condition of the structure. Therefore, the relationship between WPEI and crack status can be obtained. The results of the WPEI analysis are illustrated in Figure 13, Figure 14 and Figure 15.

Figure 13 shows WPEIs for test cases with no crack and with empty crack. As can be seen from the figure, there is a close relationship between WPEI and the length of the crack in the epoxy joint. When there is no crack in the specimen, the calculated WPEI is 22,398.1 V^2^. When the length of the empty crack is 3 cm and 5 cm, the value of WPEIs are respectively 14,152.3 V^2^ and 9031.5 V^2^, decreasing abruptly. When the length of the empty crack is 13 cm, the value of WPEI is 1119.8 V^2^, which is far less than the test case with no crack. Therefore, it is demonstrated that as the length of the empty crack increases, the value of WPEI decreases. Furthermore, when the length of the empty crack exceeds 9 cm, the descending trends of the WPEI is not obvious.

Figure 14 shows WPEIs for test cases with no crack and with water-filled crack. It is shown in the figure that there is a close relationship between WPEI and the length of the water-filled crack in the epoxy joint. When there is no crack in the specimen, the calculated WPEI is 22,398.1 V^2^. When the length of the water-filled crack is 3 cm and 5 cm, the value of WPEIs are respectively 17,982.4 V^2^ and 12,771.6 V^2^, decreasing abruptly. When the length of the empty crack is 11 cm and 13 cm, the values of WPEIs are 2443.1 V^2^ and 1531.1 V^2^, respectively. Therefore, it is demonstrated that as the length of the empty crack increases, the value of WPEI decreases. Moreover, the attenuation trend of the calculated WPEIs of the water-filled crack is consistent with that of the empty crack. 

Figure 15 shows WPEIs for all test cases to compare the difference between empty cracks and water-filled cracks. As demonstrated in the figure, the calculated WPEI for the test case with no crack is the maximum value. Because there is no crack in the epoxy joint and the stress wave can directly transmit through the joint, the SA2 can collect the most signals. The crack in the epoxy joint attenuates stress wave energy on the wave path, and the received energy is smaller than the test case with no crack. It also can be further found from the figure that, for a certain length of crack filled with water functioning as a conduit, WPEI has a certain degree of growth, compared with the empty crack. However, when the length of the crack exceeds 9 cm, the increasing trend of the WPEI is not obvious.

### 4.4. Analysis and Discussion

Based on the time-domain analysis, frequency-domain analysis and WPEI analysis, it can be seen that the attenuation trend of the tested amplitude is consistent with the length of the crack. When there is a crack in the epoxy joint, the amplitudes decrease significantly. When the length of the crack is more than 9 cm, the changes in the amplitudes are relatively small. It also can be further figured out that, for a certain length of the crack, the tested amplitude of the water-filled crack is larger than the empty crack. When the length of the crack is more than 9 cm, the increasing trend of the amplitudes is relatively small. This paper develops the relationship between the length of the crack and the propagation of the stress wave. Moreover, the WPEI results allow us to distinguish the crack without and with water if the crack length is known. Therefore, the crack and the water in the crack of the epoxy joint of PCSB can be detected and monitored using piezoceramic-based SA with an active sensing method.

It should be noted that there is much future research work to do to monitor the joint quality in a real scenario without the prior knowledge of the length of the crack and the presence of water. In future research, we will try to propose a new index, optimize the arrangement of SAs, and conduct a theoretical study on the physical mechanism of stress wave propagation in water-filled cracks, to evaluate the dimension of a crack or a leakage in real PCSBs.

## 5. Conclusions

A precast concrete segmental bridge (PCSB) can be rapidly constructed with the assembly method and the structural health of this type of bridge largely depends on the performance of the joint between the assembled segments. The crack and leakage in the epoxy joint will affect the serviceability and durability of the structure. A proposed method for detecting the joint quality and leakage for PCSB with embedded smart aggregate (SA) based on the active sensing method is developed. The feasibility of this developed method was studied by an assembled specimen with different lengths of empty cracks and water-filled cracks. Time-domain analysis, frequency-domain analysis and wavelet-packet-based energy index (WPEI) analysis were conducted to evaluate the health condition of the structure. It is demonstrated that as the length of the empty crack increases, the received aptitude, Power Spectrum Density (PSD) energy and WPEI decrease. For a certain length of the water-filled crack, compared with the empty one, the received aptitude, PSD energy and wavelet-packet-based energy index (WPEI) have a certain degree of growth. Furthermore, as the length of the water filled crack increases, the value of the received aptitude, PSD energy and WPEI decrease. By comparing the collected voltage signals, PSD energy and WPEI under different conditions, the test results are closely related to the length of the crack and leakage in the epoxy joint of PCSB. In conclusion, this paper develops the relationship between the length of the crack and the propagation of the stress wave. Moreover, the WPEI results allow us to distinguish the crack without and with water if the crack length is known. Therefore, the devised approach has certain application value in detecting the crack and leakage in the joint of PCSBs.

## Figures and Tables

**Figure 1 sensors-20-05398-f001:**
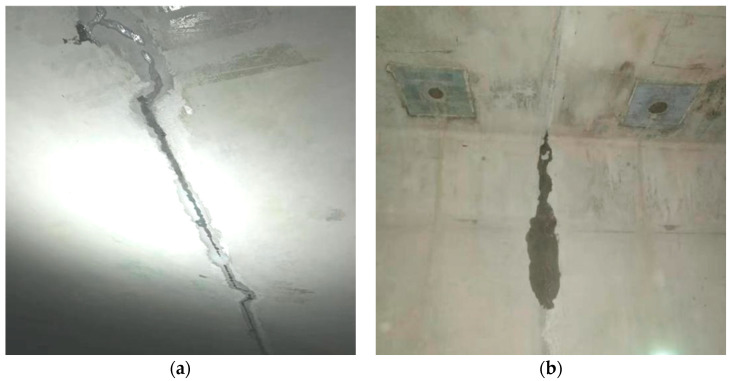
Leakage in the box girder of a newly constructed precast segmental bridge. (**a**) Leakage in the top flange; (**b**) Leakage in the web.

**Figure 2 sensors-20-05398-f002:**
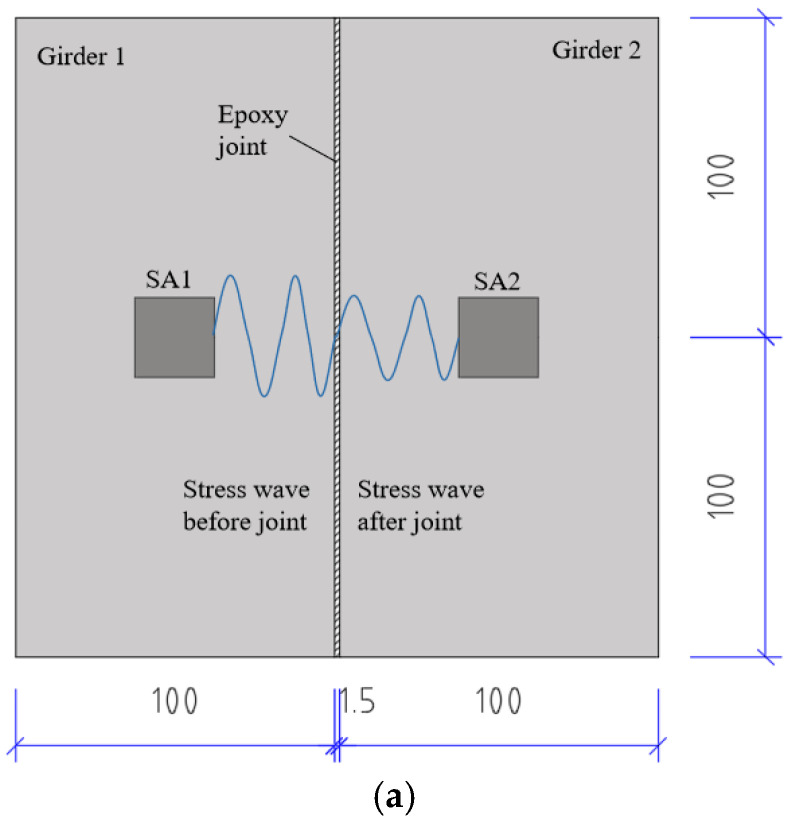
Schematic of stress wave propagating path (unit: mm). (**a**) No crack (**b**) With 3 cm crack; (**c**) With 3 cm crack filled with water.

**Figure 3 sensors-20-05398-f003:**
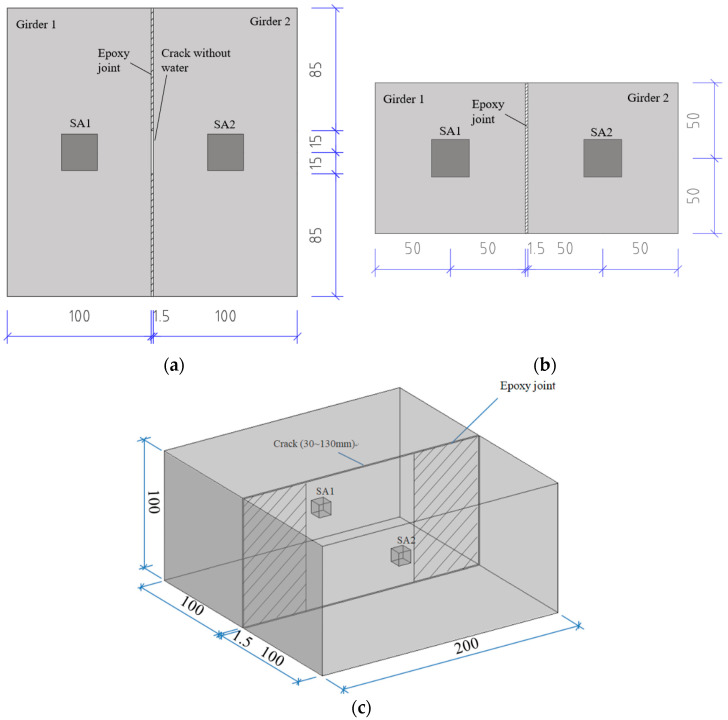
Design of the test specimen (unit: mm). (**a**) The plan view. (**b**) The elevation view. (**c**) Three-dimensional view.

**Figure 4 sensors-20-05398-f004:**
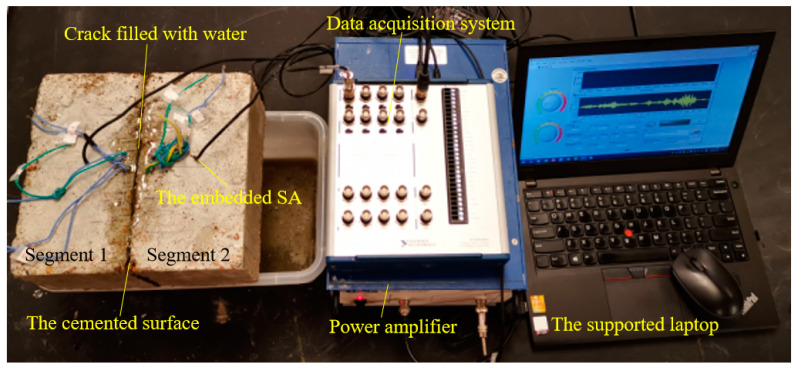
Experimental setup.

**Figure 5 sensors-20-05398-f005:**
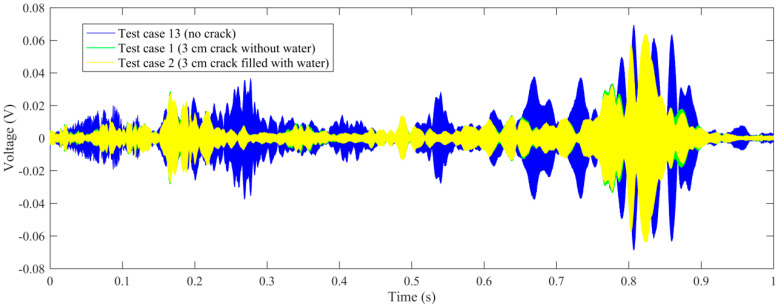
The comparison of voltage signals: test case 13 (no crack), test case 1 (3 cm crack without water), test case 2 (3 cm crack filled with water).

**Figure 6 sensors-20-05398-f006:**
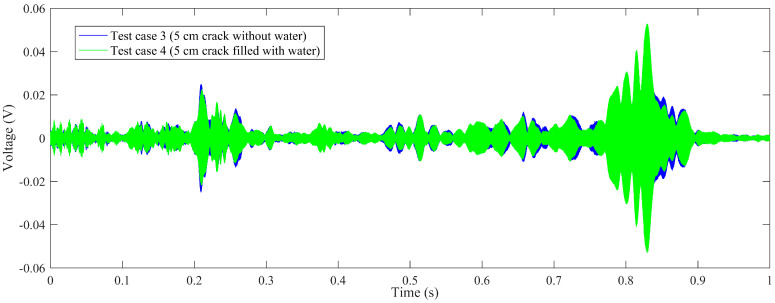
The comparison of voltage signals: test case 3 (5 cm crack without water), test case 4 (5 cm crack filled with water).

**Figure 7 sensors-20-05398-f007:**
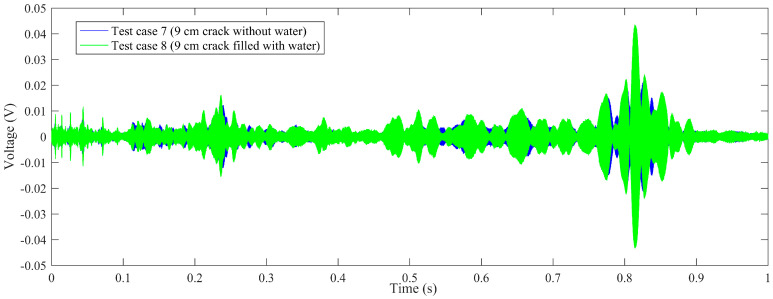
The comparison of voltage signals: test case 7 (9 cm crack without water), test case 8 (9 cm crack filled with water).

**Figure 8 sensors-20-05398-f008:**
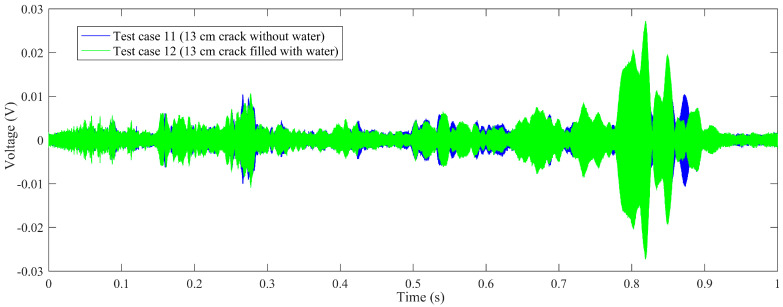
The comparison of voltage signals: test case 11 (13 cm crack without water), test case 12 (13 cm crack filled with water).

**Figure 9 sensors-20-05398-f009:**
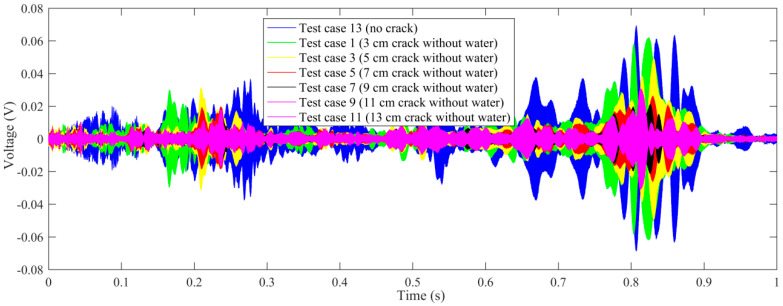
The comparison of voltage signals (test case: no crack and all cracks without water).

**Figure 10 sensors-20-05398-f010:**
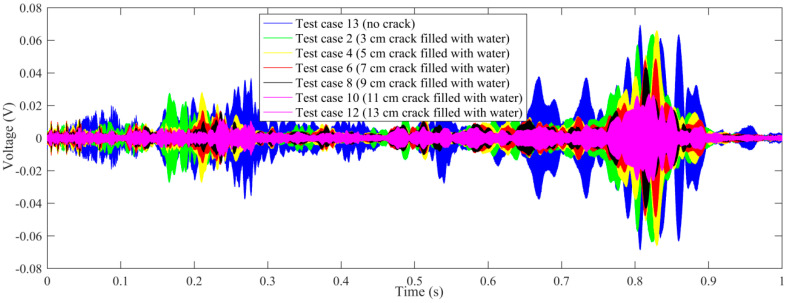
The comparison of voltage signals (test case: no crack and all cracks filled with water).

**Figure 11 sensors-20-05398-f011:**
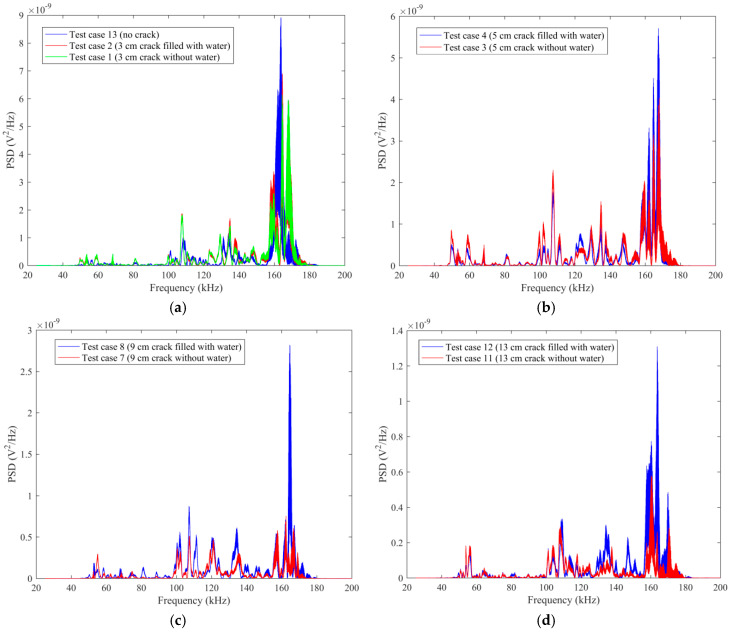
The comparison of Power Spectrum Density (PSD) energy: (**a**) Test case 13 (no crack), test case 1 (3 cm crack without water), test case 2 (3 cm crack filled with water); (**b**) Test case 3 (5 cm crack without water), test case 4 (5 cm crack filled with water); (**c**) Test case 7 (9 cm crack without water), test case 8 (9 cm crack filled with water); (**d**) Test case 11 (13 cm crack without water), test case 12 (13 cm crack filled with water).

**Figure 12 sensors-20-05398-f012:**
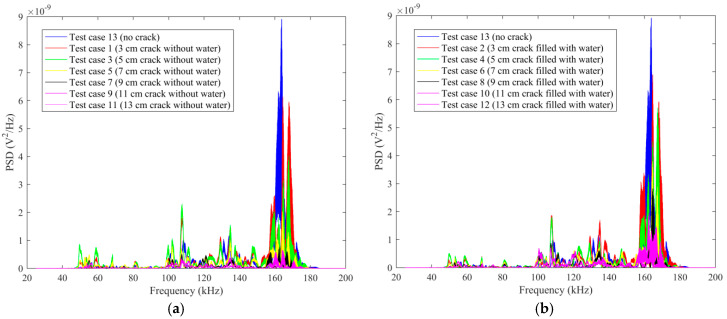
The comparison of PSD energy: (**a**) Test case: no crack and all cracks without water; (**b**) Test case: no crack and all cracks filled with water.

**Figure 13 sensors-20-05398-f013:**
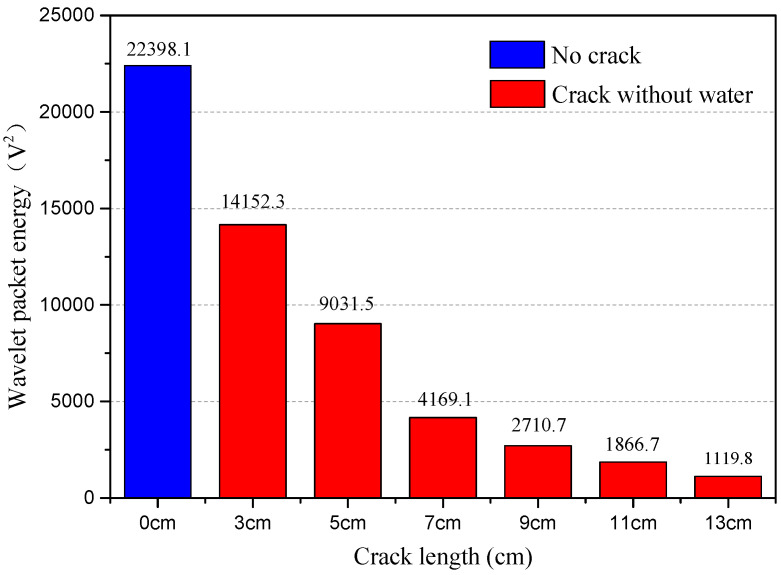
Wavelet-packet-based energy indices (WPEIs) of collected signals for test cases with no crack and empty crack.

**Figure 14 sensors-20-05398-f014:**
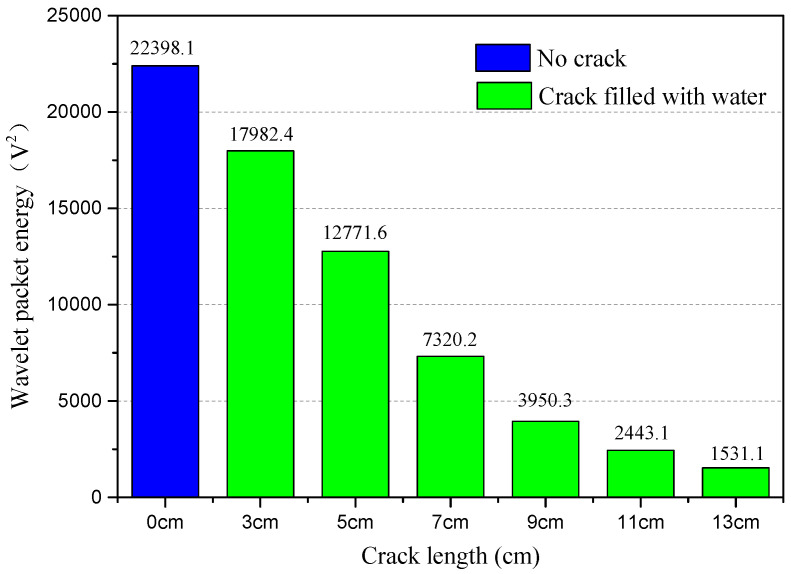
WPEIs of collected signals for test cases with no crack and water-filled crack.

**Figure 15 sensors-20-05398-f015:**
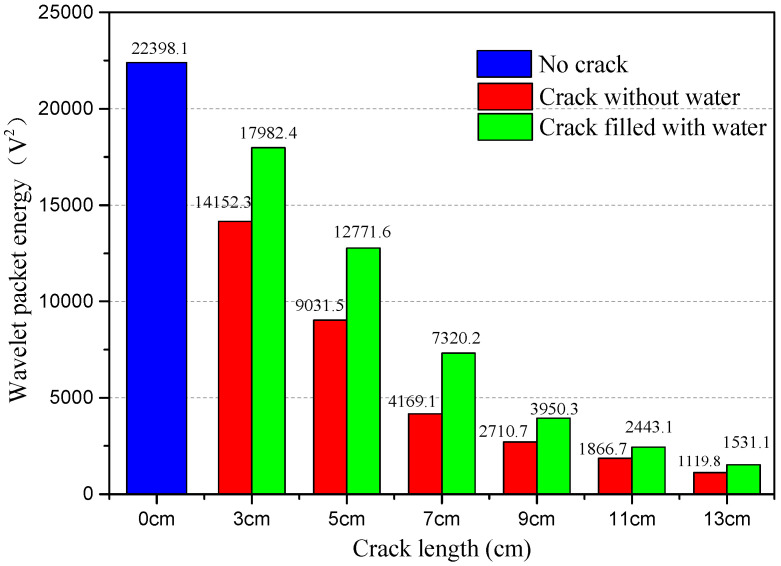
WPEIs of collected signals for all test cases.

**Table 1 sensors-20-05398-t001:** Material properties of concrete, epoxy and smart aggregate (SA).

Materials	Parameters	Value	Units
Concrete	Density	2422	kg/m^3^
	Young’s modulus	34.2	Gpa
	Poisson’s ratio	0.17	/
Epoxy	Density	1250	kg/m^3^
	Young’s modulus	3.5	Gpa
	Poisson’s ratio	0.1	/
SA	Dimension	25 × 25 × 25	mm
	Piezoelectric strain coefficients (d33)	4.00	10^−10^ C/N

**Table 2 sensors-20-05398-t002:** The 13 test cases.

Test Case	1	2	3	4	5	6	7	8	9	10	11	12	13
Crack length (cm)	3	3	5	5	7	7	9	9	11	11	13	13	0
Filled with water	N	Y	N	Y	N	Y	N	Y	N	Y	N	Y	-

**Table 3 sensors-20-05398-t003:** The detailed parameters of the swept sine wave signal.

Parameters	Initial Frequency (Hz)	Final Frequency (Hz)	Duration (s)	Amplitude (V)	Input Rate (Hz)	Onput Rate (Hz)
Value	1000	200,000	1	3	2,000,000	2,000,000

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
