# Peer review of "Detecting of the Crack and Leakage in the Joint of Precast Concrete Segmental Bridge Using Piezoceramic Based Smart Aggregate"

_sensors, 2020, doi:10.3390/s20185398_

Round 1

Reviewer 1 Report

The aim of the paper is an experimental investigation related to crack in samples (joints between concrete elements). The damage detection method based on wave propagation between two PZTs embedded into the joint.

Remarks:

1) Is there any relationship among time signals for consecutive stages of the crack elongation? The analyses in time domain are limited to maximal amplitudes of the signals. What about shape?

2) Is it possible to combine the amplitudes of chosen frequencies with crack length and the water occurrence?

3) What is a physical reason that water occurrence gives higher values of WPEIs than for empty crack? What if the gap will be filled in water partially?

4) The WPEIs results presented in fig. 14 allow to distinguish crack without and with water if the crack length is known. However, how to distinguish empty crack with length equal to 7 cm (4169.1) from crack with water and length 9 cm (3950.3) without additional information, having only the measured signal?

5) Something wrong happened with fig. 11. It should be corrected.

Author Response

We first would like to express our sincere appreciation to the Academic Editor and Assistant Editor for their time and efforts in coordinating the peer-review process, and two anonymous reviewers for their valuable comments on our manuscript. Their valuable comments helped the authors to significantly improve the quality of this manuscript. All the reviewers’ comments have been seriously taken into consideration and thoroughly addressed in the revised manuscript, and the explanation/corrections are provided in the following item-to-item response to each of the reviewers’ comments. To make the revised manuscript easier for editors and reviewers to read, all changes are printed in blue in the revised manuscript. Please see our replies to the reviewers’ comments for more details. Comments and Suggestions for Authors The aim of the paper is an experimental investigation related to crack in samples (joints between concrete elements). The damage detection method based on wave propagation between two PZTs embedded into the joint. Response: Thanks for your encouraging recommendation and valuable comments, which have received our serious consideration, and actions have been taken to address these suggestions and comments. Please see the following point-by-point replies for details. Detailed comments 1.Is there any relationship among time signals for consecutive stages of the crack elongation? The analyses in time domain are limited to maximal amplitudes of the signals. What about shape? Response: Thanks for the comment. In Section 4 of the manuscript, by comparing the collected voltage signals, PSD energy and WEPIs under different conditions, the test results are closely related to the length of the crack in the epoxy joint of PCSB. It can be inferred that, for consecutive stages of the crack elongation, as the length and the width of the crack increases, the collected voltage signals, PSD energy and WEPI decrease. It can be seen from Figures 9-10, the shape of the signal changes as the healthy state of the specimen changes. However, it is easier to quantify the healthy state with maximal amplitudes of the signals. Therefore, the analyses in time domain are mainly on maximal amplitudes of the signals. The relevant supplementary explanation has been added in the second paragraph of Subsection 4.1 of the revised manuscript. 2.Is it possible to combine the amplitudes of chosen frequencies with crack length and the water occurrence? Response: Thanks for the suggestion. In the revised manuscript, the amplitudes of PSD energy with crack length and the water occurrence are combined and discussed in detail. 3.What is a physical reason that water occurrence gives higher values of WPEIs than for empty crack? What if the gap will be filled in water partially? Response: Thanks for the comment. An empty crack in the epoxy joint will attenuate stress wave energy on the wave path. While the crack is filled with water, which functions as a wave conduit, the received energy will be more than that of empty crack. Moreover, it can be inferred when the crack is filled in water partially, the received energy will be more than that of empty crack, but less than that of crack filled in water fully. 4.The WPEIs results presented in fig. 15 allow to distinguish crack without and with water if the crack length is known. However, how to distinguish empty crack with length equal to 7 cm (4169.1) from crack with water and length 9 cm (3950.3) without additional information, having only the measured signal? Response: Thank you for your comment. The current work discusses the indices such as the collected voltage signals, PSD energy and WEPIs to evaluate the different healthy states of the specimen with different length of empty crack and water filled crack. The WPEIs results presented in Figure 15 allow to distinguish crack without and with water if the crack length is known. It should be noted that, there is much future research work to do to monitor the joint quality in a real scenario without the prior knowledge of the length of the crack and the presence of water. In future research, we will try to propose a new index for evaluating the dimension of a crack or a leakage in real PCSBs. In the abstract, Subsection 4.4 and Section 5 of the revised manuscript, the description of the conclusion has been revised. 5.Something wrong happened with fig. 11. It should be corrected. Response: Thank you for your careful observation. Figure 11 has been revised.

Reviewer 2 Report

The submitted manuscript investigates the feasibility of piezo-ceramic based smart aggregate to detect the crack and leakage between precast concrete segmental joints. Experiments have been conducted and results have been processed using time domain, frequency domain and WPEI methods. The manuscript is well written and has the merit to be published in the journal of Sensors.  Minor corrections are required to improve the quality of the paper as listed below:

  • Page 11, second paragraph, it should be Figure 11 not figure 10.

  • Please add legend/title for each subplot of figure 11, it will much easier to compare and read the figure

  • Section 4.2 needs more discussion.

In addition:

1- section 4.1 and 4.2 present the time-domain and frequency domain. It would be interesting to see the time-frequency plots for each different crack length with and without water. (Spectrogram using short-time Fourier transform)

2- please add more discussion for the FFT results, why there are different peaks and explain the change of amplitude or change in frequency in each test.

3- the main problem that the paper does not mention is that in a real scenario, not knowing the length of the crack and not knowing if there is water or not, how the results can be interpreted and can be used to find the dimension of a crack or a leakage?

4- how sensitive is the results for partial saturation of the crack.

Author Response

We first would like to express our sincere appreciation to the Academic Editor and Assistant Editor for their time and efforts in coordinating the peer-review process, and two anonymous reviewers for their valuable comments on our manuscript. Their valuable comments helped the authors to significantly improve the quality of this manuscript. All the reviewers’ comments have been seriously taken into consideration and thoroughly addressed in the revised manuscript, and the explanation/corrections are provided in the following item-to-item response to each of the reviewers’ comments. To make the revised manuscript easier for editors and reviewers to read, all changes are printed in blue in the revised manuscript. Please see our replies to the reviewers’ comments for more details. Comments and Suggestions for Authors The submitted manuscript investigates the feasibility of piezo-ceramic based smart aggregate to detect the crack and leakage between precast concrete segmental joints. Experiments have been conducted and results have been processed using time domain, frequency domain and WPEI methods. The manuscript is well written and has the merit to be published in the journal of Sensors.  Response: Thanks for your encouraging recommendation and valuable comments, which have received our serious consideration, and actions have been taken to address these suggestions and comments. Please see the following point-by-point replies for details. Detailed comments 1.Minor corrections are required to improve the quality of the paper as listed below: (1)Page 11, second paragraph, it should be Figure 11 not figure 10. Response: Thank you for your careful observation. The error has been corrected. (2) Please add legend/title for each subplot of figure 11, it will much easier to compare and read the figure Response: Thanks for the suggestion. Figure 11 has been revised. (3) Section 4.2 needs more discussion. Response: Thanks for the suggestion. In Section 4.2 of the revised manuscript, the amplitudes of frequency-domain signals with crack length and the water occurrence are combined and discussed in detail. 2. In addition: (1)section 4.1 and 4.2 present the time-domain and frequency domain. It would be interesting to see the time-frequency plots for each different crack length with and without water. (Spectrogram using short-time Fourier transform) Response: Thanks for the suggestion. In Section 4.2 of the revised manuscript, the results of frequency-domain analysis for different crack length with and without water are shown in Figure 11. In order to further study the influence of the length of the empty crack on the PSD energy, all the frequency-domain signals of the test cases with empty crack and no crack are summarized in a single figure, as shown in Figure 12(a). On the other hand, all the frequency-domain signals of the test cases with water filled crack and no crack are summarized in Figure 12(b). The relevant discussion of results are in presented section 4.2 of the revised manuscript. The linear frequency sweep is used in the experiment, in which the sweep frequency and the sweep time have a linear relationship. Therefore, we do not conduct and provide short-time Fourier transform. (2)please add more discussion for the FFT results, why there are different peaks and explain the change of amplitude or change in frequency in each test. Response: Thanks for the suggestion. In the revised manuscript, the amplitudes of frequency-domain signals with crack length and the water occurrence are combined and discussed in detail. It can be seen from Figures 11-12 that the range from 50 kHz to 180 kHz is the sensitive frequency range of the sensor signal. Moreover, the shape of the frequency-domain signals changes as the healthy state of the specimen changes. However, it is easier to quantify the healthy state using maximal PSD energy of the signals. The relationship between PSD energy and the healthy state of the specimen is indicated in Section 4.2. (3) the main problem that the paper does not mention is that in a real scenario, not knowing the length of the crack and not knowing if there is water or not, how the results can be interpreted and can be used to find the dimension of a crack or a leakage? Response: Thank you for your comment. The current work discusses the indices such as the collected voltage signals, PSD energy and WEPIs to evaluate the different healthy states of the specimen with different length of empty crack and water filled crack. The WPEIs results presented in Figure 15 allow to distinguish crack without and with water if the crack length is known. It should be noted that, there is much future research work to do to monitor the joint quality in a real scenario without the prior knowledge of the length of the crack and the presence of water. In future research, we will try to propose a new index to evaluate the dimension of a crack or a leakage in real PCSBs. In the abstract, Subsection 4.4 and Section 5 of the revised manuscript, the description of the conclusion has been revised. (4) how sensitive is the results for partial saturation of the crack. Response: Thank you for your comment. In the present work, with a total of 13 test cases, the different length of crack without water and filled with water were simulated and tested. Comparison of the collected voltage signals, PSD energy and WEPIs under different healthy states shows that, the test results are closely related to the length of the crack and the leakage in the epoxy joint. It can be inferred from the detected results that the collected voltage signals, PSD energy and WEPIs are sensitive for partial saturation of the crack.
